# Solutions to the Problem of Freight Transport Flows in Urban Logistics

**Nijolė Batarlienė *** and **Darius Bazaras**

Department of Logistics and Transport Management, Vilnius Gediminas Technical University (VILNIUS TECH), Plytinės 27, LT-10105 Vilnius, Lithuania

\* Correspondence: nijole.batarliene@vilniustech.lt; Tel.: +370-5-2370-634

**Abstract:** The operation of a city's logistics system is associated with many problematic issues, for example, intense pollution and negative impacts on the environment, uneven intensity of traffic flows, and other problems. This article focuses on heavy transport because it causes disruptions in the supply system and affects timely customer service. Optimization processes are associated with route selection, deployment of logistics centers, and the need for cargo consolidation–de-consolidation, which leads to the possibility of using the PPP (public–private partnership) method in practice. A unique aspect of this research is related to the problems of the first and last mile and the use of elements of the "sharing economy". Therefore, this article aims to analyze the problematic challenges of the first and last mile and the role and significance of heavy transport in urban logistics. For that purpose, in addition to an analysis of scientific sources, an expert survey was conducted and responses related to the questions on optimizing heavy traffic flows in city logistics were analyzed. The study data were processed using Kendall's correlation coefficient, the concordance method, and the consistency coefficient. During analysis, using the average rank transformation to weights (ARTIW) method, the subjective normalized weights of the main factors that have the greatest influence on a city's logistics system were determined. Groups of interested parties are also named separately, for whom the results of the study and the formulated decision may be relevant. Based on the results of the research, a recommendation was formulated for the development of small cargo collection and distribution services through self-service terminals located on the outskirts of cities. Proposals are presented to reorganize the system, provide details of new infrastructure elements and suburban terminals, and encourage for the use of environmentally friendly vehicles with a lower carrying capacity.

**Keywords:** urban logistics; road freight transport; traffic flows; first and last mile

## 1. Introduction

As cities grow, new challenges arise for mobility. It is estimated that 10–15% of the total mileage of a city comes from freight transport. When freight flows increase in cities, many different problems arise, such as loading and delivery delays, increasing environmental pollution and noise. Other problems also arise, such as lack of heavy transport drivers, wear and tear of roads, etc.

As the population grows, urban logistics has become the subject of scientific research and has gained great importance. In recent decades, more and more people live in cities. From 1950 to 2018, the proportion of the world's population in cities increased to 55 percent [1]. Rapid urbanization has also brought unprecedented challenges. The future sustainability of cities depends on the compatibility of development and the environment and the applied innovations [2]. Increasing traffic flows cause a number of problems [3–5]. Therefore, many researchers are looking for various solutions that can be applied in urban logistics. Dutch scientists estimate that the amount of greenhouse gas emissions across the country was as high as 0.9 million cubic meters [6].

As early as 2001, researchers [7] defined urban logistics as a process that allows complete optimization of logistics and transport activities of companies based on advanced information systems in urban areas, including traffic environment, traffic congestion, traffic safety, and energy saving, in the conditions of a market economy [8].

With the expansion of cities, access of commercial transport to the central part of a city is an increasing problem, which affects overall mobility in a city, especially during rush hours.

The main problems for the mobility of cargo traffic flow are due to outdated infrastructure, slow implementation of information technologies to create a common informational field of urban logistics, and underutilization of car capacity in order to reduce the flow of partially loaded cars in cities.

The intensity of heavy and freight transport flows in a city is uneven. The negative impact of this transport on city logistics is manifested when all traffic flows are intensive in city logistics. Freight transport negatively affects the speed of traffic flows and can be the cause of both technical accidents and traffic jams [5]. In this case, logistics companies suffer losses due to late or non-delivered cargo. The quality of service drops due to congestion.

The main function of freight transport is the physical transport of goods, including the delivery of goods to receivers within the limits of a city's logistics infrastructure [8]. Optimization of the distribution of freight transport flows can be performed by modelling the consequences of emergency situations.

The purpose of this study is to determine the subjective normalized weights of the main factors that have the greatest influence on a city's logistics system, using Kendall's correlation coefficient method and the average rank transformation of weights (ARTIW) method [9]. At the same time, the main problems of the topic under analysis are identified, along with the dependencies of the phenomena and possible trends and insights. After systematizing the obtained results, a possible solution for the optimization of freight transport flows in a city is formulated, which would provide a real basis for the optimal distribution of freight transport flows in cities.

It can be said that the study and its results are relevant for potentially interested parties, which are city municipalities, transport and logistics companies, service providers, and users located in the city boundaries. The study is universal enough and its process can be easily replicated in different cities, and the results of the existing study are applicable in different cities as well. In any case, it should be taken into account that the current study was carried out based on a city with a population of 700,000, a developed transport and logistics infrastructure and the city covers an area of 401 km$^2$. Each stakeholder group, in the context of this research, has individual objectives. These goals could be noted as follows:

- Municipalities: to reduce the burden on a city's logistics system, improve the quality of life, and optimize heavy traffic flows;
- Logistics companies: use available resources more efficiently, reduce the cost of services, reduce hazardous substances, and $CO_2$ emissions;
- Service recipients and users: to receive the necessary loads promptly and plan their activities;
- City dwellers: to live in a safe environment where traffic flows are optimized and a decent standard of living is ensured.

## 2. Materials and Methods

### 2.1. Theoretical Aspects of the Flows of Freight Road Traffic in the Urban Logistics System

These days, urban logistics planning is based on high-quality and fast cargo transportation, with various ecological solutions. Service providers must pay attention to shorter delivery times, flexibility, and service reliability [10].

Analyzing the sources of the scientific literature, it can be stated that city logistics can be seen as a tool that allows efficient management of the movement of material flow in the city territory and forms innovative solutions [11]. In most cases, the content of logistics solutions consists of analysis, planning, integrated information management, and decision making. The uniqueness of a city's logistics system is the consolidation of cargo from



various senders and receivers and the management of the cargo transportation process in a city's infrastructure [12].

Analyzing the existing terminology, it is stated that the objective of city logistics is to support and develop the harmonious development of urban areas, increase the efficiency of transport and logistics processes, and significantly reduce the negative impact on the environment [13].

The fast pace of life and the pursuit of prosperity led to the need to find new solutions for the development of cargo distribution and transportation efficiency [14]. These goals can be achieved by organizing distribution activities in a more targeted and qualitative way and by properly forming an infrastructure network, and cooperation between carrier companies is also important in this area [15]. Scientists and researchers emphasize the most important areas of urban logistics development as planning, information technology, communication, public–private partnership, support measures, and regulations [16].

Heavy transport causes many problems in cities. These include traffic congestion, air and noise pollution, and traffic accidents [17]. However, freight transport is an inevitable part of city life, as trade takes place in cities and various services are provided.

In their publications, scientists and researchers have widely analyzed the problems of the first and last mile and their practical expression forms, which depend on the complexity of the city infrastructure, the location of the terminals, and their distance to the city border [18]. The last mile is defined as the part of the logistics supply chain from the cargo terminal to the final destination of the shipment (cargo) [19].

Scientific works emphasize that to optimize the last mile, it is necessary to harmonize different solutions related to the operation of cargo terminals and warehouses, as well as transport solutions in the logistic environment of a city. However, new transport solutions are associated with the transformation of a city's logistics infrastructure, such as a network of charging stations for the use of electric cars [20]. The social aspect of the problem should also be noted, because cities have a significant impact on social and economic development, as well as an impact on the environment in general. Therefore, when analyzing city problems, it is necessary to emphasize topics related to the appropriate use of resources, optimization of the transport system, information and data management, social change, the impact on the environment, and conservation of nature [21,22].

According to researchers, great attention is paid to innovations in the field of information and communication technologies in modern city logistics:

- "Green technologies" allow for a more responsible use of a city's main resources (gas, water, $CO_2$ emissions, and introducing electric transport) and the development of renewable energy sources;
- Implementation of broadband internet 5G, which ensures the development of modern information technologies and the development of closed and secure networks;
- Artificial intelligence that makes certain decisions;
- The spread of eSim technology to control the movement of goods from the production process to sales [20,21].

Another researcher [22] studied the impact of freight transport on $CO_2$ emissions. In their research, they relied on data from the French Environment and Energy Agency (ADEME). Table 1 shows a table of ADEME data showing the distribution of $CO_2$ emissions by weight of freight cars.

**Table 1.** $CO_2$ emissions of different types of freight vehicles [22].

| Cars by Weight | $CO_2$ Emission (g/tkm) |
| --- | --- |
| Small commercial vehicle (1.5–3.5 t) | 1103 |
| Cars (6.1–10.9 t) | 435 |
| Cars (11–21 t) | 221 |
| Cars (21–32.6 t) | 196 |

A total of 82% of the cargo was also found to reach the city by low-capacity transport; the rest is transported by heavy duty freight transport [22].

The examples presented show that freight transport is unevenly distributed in a city logistics system. The use of freight transport in urban logistics is due more to the capacity and legal regulations in the urban area. In a city's logistics system, there are possible restrictions on the movement of goods and heavy vehicles and restrictions on driving on weekends and during appropriate hours. The load on the axle may also be limited, taking into account the existing state of a city's infrastructure, such as street surfaces, bridges, historical heritage, etc. City logistics is dominated by freight transport with a carrying capacity of up to 3.5 t, but the use of heavy-duty transport is also unavoidable, causing the most problems, especially during peak hours, of possible traffic jams, noise, and increased emissions due to braking, increasing traffic jams or the number of controlled traffic light intersections.

Researchers [21,23] have proposed the application of a two-tier hyperconnected urban logistics system, which, according to the authors, would help reduce heavy traffic in cities. This proposed optimization model is based on the idea that cargo from various customers could be consolidated in terminals near cities and transported by heavy transport to small distribution centers in urban areas, from where it would be distributed to recipients by urban commercial transport.

According to the researchers' proposal, large terminals in the countryside could be connected with smaller terminals in urban areas, to which transport transporting consolidated cargo would go, and distribution would be carried out by city-friendly, ecological, and small-capacity urban freight transport. According to the researchers, this would reduce heavy traffic in cities [23].

Another way to optimize transport flows in a city's logistics system is the "sharing economy". In this context, it is a very broadly interpreted concept, as it can cover both the sharing of vehicles and serviced routes. Such an "exchange" would enable transportation and logistics companies to transport more cargo faster in a more environmentally friendly way at a lower cost. Favorable conditions would be created for better utilization of resources, optimization of routes, simplification of schedules, and reduction in carbon dioxide emissions. It should be noted that, in addition to greater operational efficiency, the sharing of available and underutilized resources would allow solving problems related to the overloading of the transport system and the lack of drivers. In the area of warehousing, the sharing economy aims to increase the utilization of existing shared customer warehouses. Finally, the sharing economy presents new and creative ways to start a business and increase efficiency through on-demand staffing models and shared logistics services [24].

The distribution of freight transport in a city depends largely on the characteristics of the traffic network, for example, the road category, number of movement lanes, organization and regulation of movement, organization of stopping places, parking capacity, etc. These parameters affect the structure of vehicles in the realization of goods flows, the intensity of traffic congestion by commercial vehicles, route planning, duration of loading–unloading operations, operational costs of carriers and service providers, etc. [25].

Thus, summarizing the opinions of different researchers, it can be stated that the distribution of freight transport flows in cities is mainly influenced by the geographical location of the city, the location of industrial and commercial places, and the locations of distribution centers in the territory and periphery. Furthermore, the mobility of freight transport in a city is influenced by road infrastructure, traffic regulations and restrictions, and parking capacity. The solution to the problem of improving the distribution of freight transport flows in urban logistics could be the optimization of these transport flows in the first and last mile stages, eliminating heavy transport, and using self-service terminals in rural logistics centers and ecological transport for the collection or distribution of small loads.

*2.2. Research Methodology*

After analyzing the examined studies and the proposed solutions for the optimization of cargo transport flows, it is clear that this is relevant today and will be relevant in the future. The problems of city logistics, which are related to transport, life in cities, and environmental protection, are solved at various levels, from the strategic level to the cooperation of municipal administrations and business entities.

A survey of experts was conducted in the feasibility study to optimize cargo flows by road transport. An in-depth interview or other method can be an alternative to a survey, but a survey is more acceptable to both respondents and interviewers. This study aims to gain a deeper understanding of the studied phenomena and to determine the possibilities of optimizing freight transport and how these possibilities can be practically applied in real life. The survey was conducted according to a rated scale of questions [26]. Questions are present as statements or criteria and then ranked.

In the survey, ten respondents were interviewed, who were logistics experts who have a university degree—at least a bachelor's degree—and at least 5 years of experience in the field of international cargo transportation and cargo distribution city logistics. Experts with an average logistics experience of almost 14 years were selected.

The survey was conducted in May–June 2022. The respondents of the survey were experts in the Lithuanian logistics sector and heads of companies or company divisions. The survey was sent to respondents electronically via ref [27]. The apklausa.lt system was designed to create and conduct online surveys with unlimited quantities of questionnaires and questions and distribute it to respondents [28]. Answers to questionnaires are provided in a simple, understandable form. The results can be saved in a file that can be opened with popular office applications (OO Calc, MS Excel, or SPSS).

After evaluating the provisions and recommendations of multi-criteria decision making (MCDM) methods and concepts [29], the number of experts was determined. This number must be greater than the number of multiple-choice questions. In the case of our study, there were ten experts and the number of optional answers is eight, so it can be said that the number of experts was appropriate.

During the research, selected experts were asked questions aimed at obtaining information on city logistics, environmental protection and environmental impact, freight traffic flows, traffic throughput, and infrastructure impact. At the same time, an attempt was made to obtain the prognostic information required for the study on possible optimization measures and solutions, possible financial aspects of the use of electricity or other ecological transport, and the use of information technologies.

The data from the expert survey, expressed in numerical values, were analyzed and calculated for the reliability of the data. The processing of the data was aimed at obtaining generalized data to study the possibilities of optimizing the flow of freight transport in a city logistics system.

Research data were processed using Kendall's concordance coefficient method of agreement [9] and the average rank transformation to weights (ARTIW) method [30,31]. All expert assessments were ranked.

Kendall's coefficient of concordance was applied to calculate the survey results and determine the distribution and concordance of opinions. The values of the concordance coefficient (W) are in the range from 0 to 1. The higher the obtained value (W), the less the opinions differ and are aligned.

To find Kendall's concordance coefficient, the sum of the ranks, $R_j$, assigned to each *j*-th criterion by *n* experts is calculated according to the formula:

$$R_j = \sum_{i=1}^{n} R_{ij} (j = 1, 2, \ldots m). \tag{1}$$

More precisely, it is based on the sum, $S$, of the squared deviations, $R_j$ (the variance analogue), from the mean rank, $\overline{R}$:

$$S = \sum_{j=1}^{m} \left( R_j - \overline{R} \right)^2. \tag{2}$$

The following calculates the average criterion rank, $\overline{R}$. It is obtained by dividing the sum of the ranks, assigned to the criterion by the experts, by the number of criteria, $m$:

$$\overline{R} = \frac{\sum_{j=1}^{m} R_{ij}}{m} = \frac{\sum_{i=1}^{n} \sum_{j=1}^{m} R_{ij}}{m}, \tag{3}$$

where $R_{ij}$ is the rank assigned by the $i$-th expert to the $j$-th criterion; $n$ is the number of experts ($i = 1, 2, \ldots, n$); and $m$ is the number of criteria ($j = 1, 2, \ldots, m$).

When $S$ (true sum of squared values) is calculated according to the formula (2), then the correlation coefficient, $W$, can be calculated according to the formula:

$$W = \frac{12 \cdot S}{n^2 \cdot (m^3 - m)} \tag{4}$$

When the opinions of the experts are similar, the value of the concordance coefficient, $W$ is about one, and if the opinions differ greatly, the value of $W$ is about zero.

The value of the random number $S$ is calculated by summing the squared values of all criteria enclosed in square brackets (Formula (5)).

$$S = \sum_{j=1}^{m} \left[ \sum_{i=1}^{n} R_{ij} - \frac{1}{2} n \cdot (m+1) \right]^2, \tag{5}$$

where $m$ is the number of criteria ($j = 1, 2, ..., m$) and $n$ is the number of experts ($i = 1, 2, ..., n$).

The following rule has been proven [30]: when the number of criteria is $m > 7$, the significance of the correlation coefficient, $W$, can be determined using Pearson's criteria (chi-square test). The random value is distributed according to $\chi^2$:

$$\chi^2 = n \cdot (m-1) \cdot W = \frac{12 \cdot S}{n \cdot m \cdot (m+1)}, \tag{6}$$

with the degree of freedom $\nu = m - 1$. Based on the selected confidence level $\alpha$ (which is assumed to be 0.05 or 0.01), the critical value $\chi^2{}_{\nu,\alpha}$ is found from the table of $\chi^2$ distribution with the degree of freedom $\nu = m - 1$. If the value of $\chi^2$ calculated by formula (6) is larger than $\chi^2{}_{\nu,\alpha}$, it shows that the experts' estimates are consistent.

The smallest value of the concordance coefficient, $W_{\min}$, can be estimated by applying the following formula (7):

$$W_{\min} = \frac{\chi^2{}_{\nu,\alpha}}{n \cdot (m-1)} \tag{7}$$

where $n$ is the expert opinions; $m$ is the number of comparative criteria that indicate the quality of an object under analysis with the selected levels of significance $\alpha$, and the degree of freedom $\nu = m - 1$. Having calculated this value, if it is not possible to assert that the experts' opinions are in agreement, $\chi^2{}_{\nu,\alpha}$ (the critical Pearson's statistic) at the degree of freedom and significant level are taken [30].

Next, the criteria of the ARTIW method [31], which describe the significances (weights), are calculated. For that, the following Formula (8) is applied:

$$\omega_j = \frac{(m+1) - \overline{R}_j}{\sum\limits_{j=1}^{m} \overline{R}_j} \tag{8}$$

where $m$ is the number of criteria ($j$ = 1, 2,..., $m$) and $\overline{R}_j$ is the average rank of the $j$-th criterion calculated according to Formula (9):

$$\overline{R}_j = \frac{\sum\limits_{i=1}^{n} R_{ij}}{n} (j = 1, 2, \ldots m), \tag{9}$$

where $R_{ij}$ is the rank of the criteria granted by the experts and $n$ is the number of experts.

## 3. Results

During the research, 10 respondents, who were experts in the field of logistics with at least 5 years of experience in international cargo transportation and cargo consolidation, as well as distribution in urban logistics, were interviewed. All experts who participated in the survey have a higher education degree, and their total average work experience in the field of logistics is almost 14 years.

Experts assessed the influence of each factor on a scale of eight points (1—the least influence; 8—the greatest influence). Answers could not be repeated.

The following factors were presented for the survey:

A1—Urban road infrastructure;
A2—Traffic regulations;
A3—Legal regulations;
A4—Geographical location of the city;
A5—Geographical distribution of economic operators;
A6—Innovation and information technology;
A7—Cooperation between economic operators;
A8—Cooperation between the city administration and economic operators.

The ratings provided by each expert for each factor were summed. The responses' evaluations were converted into ranks according to the formula $R_{ij} = (n + 1) - B_{ij}$ and listed in Table 2.

**Table 2.** Calculations of the factors that have the greatest impact on a city's logistics system.

| Codes of Experts | Factors That Have the Greatest Impact on a City's Logistics System ($j$ = 1, 2, . . . , 8) | | | | | | | |
|---|---|---|---|---|---|---|---|---|
| | A1 | A2 | A3 | A4 | A5 | A6 | A7 | A8 |
| E1 | 3 | 1 | 7 | 2 | 5 | 6 | 8 | 4 |
| E2 | 1 | 2 | 3 | 4 | 5 | 6 | 7 | 8 |
| E3 | 2 | 1 | 8 | 3 | 6 | 7 | 5 | 4 |
| E4 | 1 | 2 | 3 | 8 | 7 | 6 | 5 | 4 |
| E5 | 1 | 3 | 8 | 2 | 7 | 6 | 5 | 4 |
| E6 | 1 | 2 | 6 | 7 | 8 | 4 | 5 | 3 |
| E7 | 1 | 2 | 7 | 4 | 5 | 8 | 6 | 3 |
| E8 | 2 | 3 | 4 | 5 | 6 | 8 | 6 | 7 |
| E9 | 4 | 5 | 7 | 1 | 2 | 8 | 6 | 3 |

**Table 2.** *Cont.*

| Codes of Experts | Factors That Have the Greatest Impact on a City's Logistics System ($j = 1, 2, \ldots, 8$) | | | | | | | |
| --- | --- | --- | --- | --- | --- | --- | --- | --- |
| | A1 | A2 | A3 | A4 | A5 | A6 | A7 | A8 |
| E10 | 2 | 4 | 7 | 1 | 8 | 3 | 6 | 5 |
| Sum of the Ranks $R_j = \sum\limits_{i=1}^{n} R_{ij} (j = 1, 2, \ldots m).$ | 18 | 25 | 60 | 37 | 59 | 62 | 59 | 45 |
| Average Rank $\bar{R} = \dfrac{\sum\limits_{j=1}^{m} R_{ij}}{m} = \dfrac{\sum\limits_{i=1}^{n}\sum\limits_{j=1}^{m} R_{ij}}{m},$ | 1.8 | 2.5 | 6 | 3.7 | 5.9 | 6.2 | 5.9 | 4.5 |
| Subtraction of the sum of ranks and a constant $\sum\limits_{i=1}^{n} R_{ij} - \frac{1}{2}n(m+1)$ | −27 | −20 | 15 | −8 | 14 | 17 | 14 | 0 |
| $\left[\sum\limits_{i=1}^{n} R_{ij} - \frac{1}{2}n(m+1)\right]^2$ | 729 | 400 | 225 | 64 | 196 | 289 | 196 | 0 |

Kendall's concordance coefficient was calculated according to Equation (4).

$W = 12 \times 2099/100 \times (512 - 8) = 0.500$, since the obtained number is equal to the value of 0.5, it can be said that the opinions of the respondents are aligned.

The weight of the concordance coefficient was calculated according to Equation (6):

$\chi^2 = 12 \times 2099/10 \times 8(8 + 1) = 34.98$. Since the calculated value of $\chi^2$ is greater than the value of $\chi^2_{kr}$ (18.4753), the opinion of the respondents is considered to be consistent and the average ranks show a common opinion.

According to Equation (7), the lowest value of the concordance coefficient, $W_{min}$, was calculated:

$W_{min} = 18.4753/10 \, (8 - 1) = 0.264 << 0.500$.

The calculations showed that the opinions of ten respondents agree on eight factors that influence the optimization of a city's logistics system, and the opinions of all respondents are summarized.

Next, the indicators of the importance of factors influencing the mobility of urban logistics and ecology, $Q_j$, were calculated according to the ARTIW method.

The correlation of the criteria is calculated according to the equation:

$$\bar{q} = \frac{\bar{R}_j}{\sum\limits_{j=1}^{m} R_J}. \tag{10}$$

The inverse is calculated using the equation $d_j = 1 - \bar{q}_j = 1 - \dfrac{\bar{R}_j}{\sum\limits_{j=1}^{m} R_j}.$

A weight indicator was obtained using this equation $Q_j = \dfrac{d_j}{\sum\limits_{j=1}^{m} d_j} = \dfrac{d_j}{m-1}.$

Indicators according to all criteria were calculated in a similar sequence and are given in Table 3.

According to the equation $\widetilde{Q}_j = \dfrac{\sum\limits_{i=1}^{n} B_{ij}}{\sum\limits_{i=1}^{n}\sum\limits_{j=1}^{m} B_{ij}}$, the indicators of the importance of each factor were calculated and are also summarized in Table 3.

The distribution of factors that influence the mobility and ecology according to importance is shown graphically in Figure 1.

**Table 3.** Distribution of factors according to importance.

| Indicators | A1 | A2 | A3 | A4 | A5 | A6 | A7 | A8 |
|---|---|---|---|---|---|---|---|---|
| $\bar{q}_j$ | 0.049 | 0.068 | 0.164 | 0.101 | 0.162 | 0.170 | 0.162 | 0.123 |
| $d_j$ | 0.951 | 0.932 | 0.836 | 0.899 | 0.838 | 0.830 | 0.838 | 0.877 |
| $Q_j$ | 0.136 | 0.133 | 0.119 | 0.128 | 0.120 | 0.119 | 0.120 | 0.125 |
| $\tilde{Q}_j$ | 0.200 | 0.181 | 0.083 | 0.147 | 0.086 | 0.078 | 0.086 | 0.125 |
| Importance | 1 | 2 | 7 | 3 | 5 | 8 | 6 | 4 |

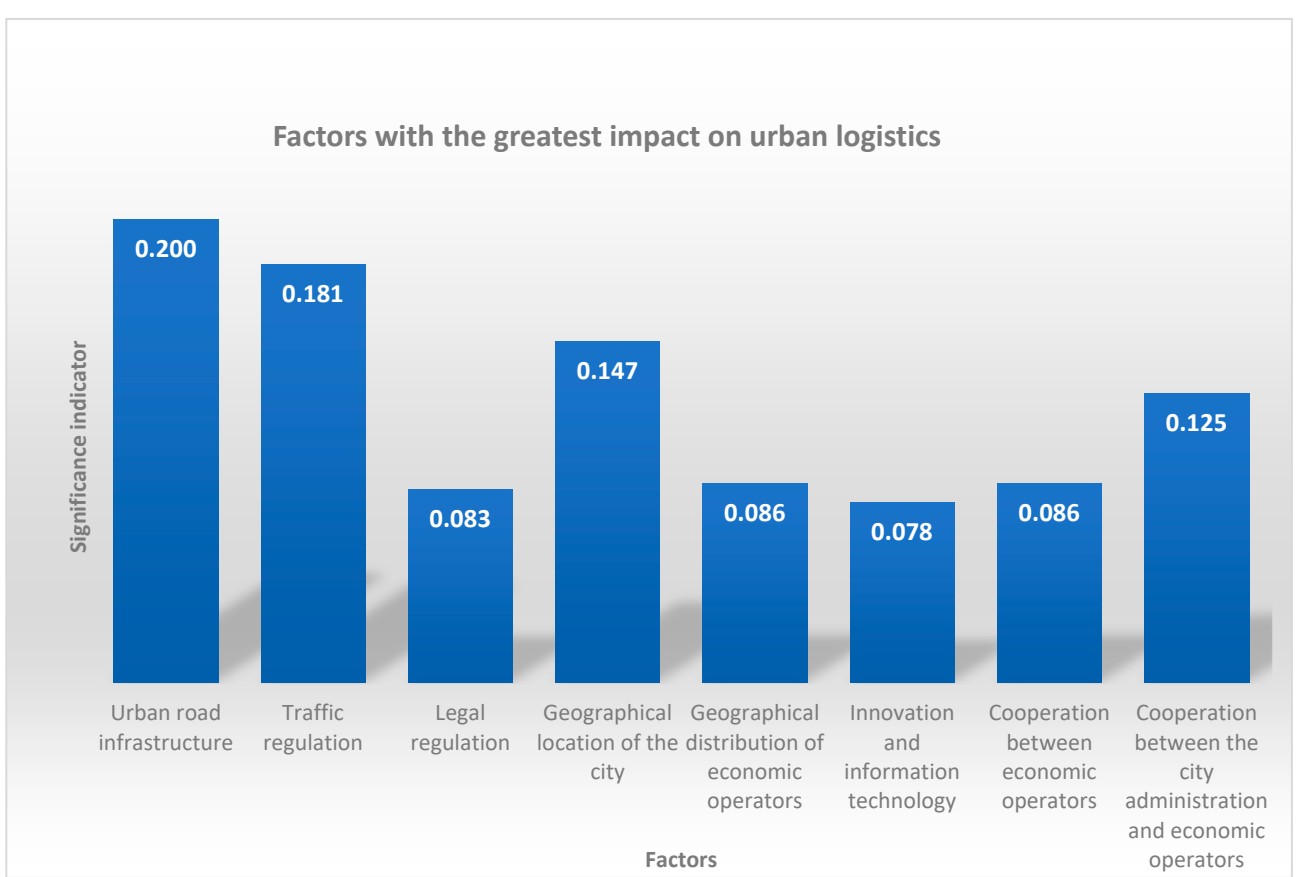

**Figure 1.** Factors that have the greatest impact on a city's logistics system.

After calculating the importance indicators, it can be concluded that legal regulations, innovation and information technology, geographic location of economic entities, and cooperation of economic entities currently have a smaller influence on city logistics (importance indicators of 0.083, 0.078, 0.086, and 0.086, respectively). On the contrary, according to experts, traffic regulation and city road infrastructure have the greatest influence on city logistics (indicators of 0.181 and 0.2, respectively).

The same methodology was used for the calculations.

The ratings provided by each expert for each factor were summed. Respondents' evaluations were converted into ranks and listed in Table 4.

**Table 4.** Calculations of factors that have the greatest influence on the optimization of a city's logistics system.

| Codes of Experts | Factors That Have the Greatest Influence on the Optimization of a City Logistics System ($j = 1, 2, \ldots, 8$) | | | | | | | |
|---|---|---|---|---|---|---|---|---|
| | **B1** | **B2** | **B3** | **B4** | **B5** | **B6** | **B7** | **B8** |
| E1 | 3 | 1 | 5 | 4 | 8 | 7 | 2 | 6 |
| E2 | 1 | 2 | 4 | 6 | 8 | 5 | 7 | 3 |
| E3 | 2 | 3 | 1 | 8 | 5 | 7 | 6 | 4 |
| E4 | 1 | 3 | 2 | 8 | 6 | 5 | 7 | 4 |
| E5 | 1 | 2 | 8 | 3 | 6 | 4 | 7 | 5 |
| E6 | 2 | 3 | 1 | 4 | 6 | 8 | 7 | 5 |
| E7 | 1 | 2 | 3 | 5 | 7 | 8 | 6 | 4 |
| E8 | 1 | 4 | 7 | 6 | 8 | 5 | 3 | 2 |
| E9 | 5 | 6 | 3 | 7 | 8 | 2 | 4 | 1 |
| E10 | 1 | 2 | 4 | 5 | 7 | 8 | 6 | 3 |
| Sum of the Ranks $R_j = \sum\limits_{i=1}^{n} R_{ij} (j = 1, 2, \ldots m).$ | 18 | 28 | 38 | 56 | 69 | 59 | 55 | 37 |
| Average Rank $\overline{R} = \dfrac{\sum\limits_{j=1}^{m} R_{ij}}{m} = \dfrac{\sum\limits_{i=1}^{n}\sum\limits_{j=1}^{m} R_{ij}}{m},$ | 1.8 | 2.8 | 3.8 | 5.6 | 6.9 | 5.9 | 5.5 | 3.7 |
| Subtraction of the sum of ranks and a constant $\sum\limits_{i=1}^{n} R_{ij} - \frac{1}{2}n(m+1)$ | −27 | −17 | −7 | 11 | 24 | 14 | 10 | −8 |
| $\left[\sum\limits_{i=1}^{n} R_{ij} - \frac{1}{2}n(m+1)\right]^2$ | 729 | 289 | 49 | 121 | 576 | 196 | 100 | 64 |

The following factors were presented for the survey:

B1—Prohibition of heavy transport entering the central part of a city;

B2—Ecological fee for freight transport in the city;

B3—State promotion for the purchase of environmentally friendly vehicles;

B4—Relocation of large economic entities to the periphery of the city;

B5—Promotion of the state by implementing information technology and connecting to the smart city system;

B6—Tax incentives for business entities cooperating with each other for the transportation of goods in the city territory;

B7 Connecting logistics companies to a unified information system for city cargo transportation;

B8—Creation of a network of small self-service terminals on the outskirts of the city.

Kendall's concordance coefficient was calculated according to Formula (4).

W = 12 × 2124/100 × (512 − 8) = 0.506, and since the number obtained is greater than the value of 0.5, it can be said that the opinions of the respondents are aligned.

The weight of the concordance coefficient was calculated according to Equation (6):

$\chi^2$ = 12 × 2124/10 × 8(8 + 1) = 35.40. Since the calculated value of $\chi^2$ is greater than the value of $\chi^2_{kr}$ (18.4753), the opinion of the respondents is considered to be consistent and the average ranks show a common opinion.

According to Formula (7), the lowest value of the concordance coefficient, $W_{min}$, was calculated:

$W_{min}$ = 18.4753/10(8 − 1) = 0.264 << 0.506

The results of these calculations are presented in Table 4.

The calculations showed that the opinions of the ten respondents agree on eight factors that influence the optimization of a city's logistics system, and the opinions of all respondents are summarized.

According to the same methodology, indicators of the importance, $Q_j$, of factors influencing the optimization of a city's logistics system were calculated.

The normalized subjective weights of the main factors with the greatest influence on the optimization of a city logistics system were calculated in the same way as the previous calculations and are summarized in Table 5.

**Table 5.** Distribution of indicators according to importance.

| Indicators | B1 | B2 | B3 | B4 | B5 | B6 | B7 | B8 |
|---|---|---|---|---|---|---|---|---|
| $\bar{q}_j$ | 0.050 | 0.078 | 0.106 | 0.156 | 0.192 | 0.164 | 0.153 | 0.103 |
| $d_j$ | 0.950 | 0.922 | 0.894 | 0.844 | 0.808 | 0.836 | 0.847 | 0.897 |
| $Q_j$ | 0.136 | 0.132 | 0.128 | 0.121 | 0.115 | 0.119 | 0.121 | 0.128 |
| $\widetilde{Q}_j$ | 0.200 | 0.172 | 0.144 | 0.094 | 0.058 | 0.086 | 0.097 | 0.147 |
| Importance | 1 | 2 | 4 | 6 | 8 | 7 | 6 | 3 |

The obtained results show that when optimizing s city's logistics system, it is necessary to take into account the ban on entering the city center (indicator 0.2) and the ecological tax (indicator 0.172), because these indicators have the greatest importance. The second most important indicators are the creation of a network of self-service terminals on the outskirts of the city (index 0.147) and support for the purchase of environmentally friendly vehicles (index 0.144). The third is the integration of logistics companies into a unified one, followed by an urban freight transport information system (index 0.097) (Figure 2). The results obtained from the research show that the role of the state is strategically important in optimizing freight transport flows in a city logistics system.

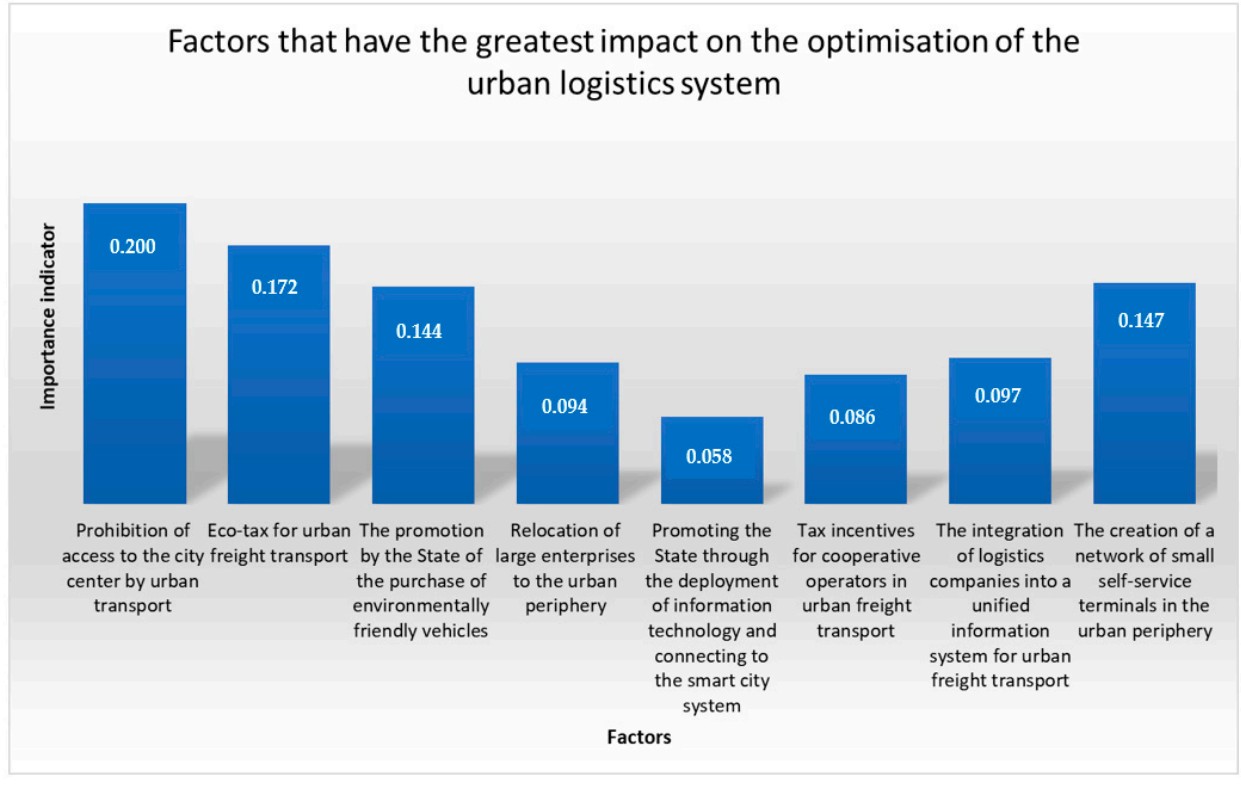

**Figure 2.** Factors that have the greatest impact on the optimization of an urban logistics system.

The study also aimed to assess which cargo vehicles in urban logistics pose the greatest problem of mobility and urban ecology, taking into account the carrying capacity of the vehicles. Respondents rated the impact of each vehicle on a scale of eight points (1—the least impact; 8—the greatest impact).

The indicators according to all criteria were calculated in a similar sequence and are shown in Table 6.

**Table 6.** Distribution of factors according to importance.

| Indicators | Up to 999 | 1000–1499 | 1500–2999 | 3000–4999 | 5000–6999 | 7000–9999 | 10,000–14,999 | 15,000 and Above |
|---|---|---|---|---|---|---|---|---|
| $\overline{q}_j$ | 0.189 | 0.175 | 0.153 | 0.125 | 0.122 | 0.117 | 0.069 | 0.050 |
| $d_j$ | 0.811 | 0.825 | 0.847 | 0.875 | 0.878 | 0.883 | 0.931 | 0.950 |
| $Q_j$ | 0.116 | 0.118 | 0.121 | 0.125 | 0.125 | 0.126 | 0.133 | 0.136 |
| $\widetilde{Q}_j$ | 0.061 | 0.075 | 0.097 | 0.125 | 0.128 | 0.133 | 0.181 | 0.200 |
| Importance | 8 | 7 | 6 | 5 | 4 | 3 | 2 | 1 |

The distribution of factors that influence the infrastructure and the entire city's logistics system by importance is shown graphically in Figure 3.

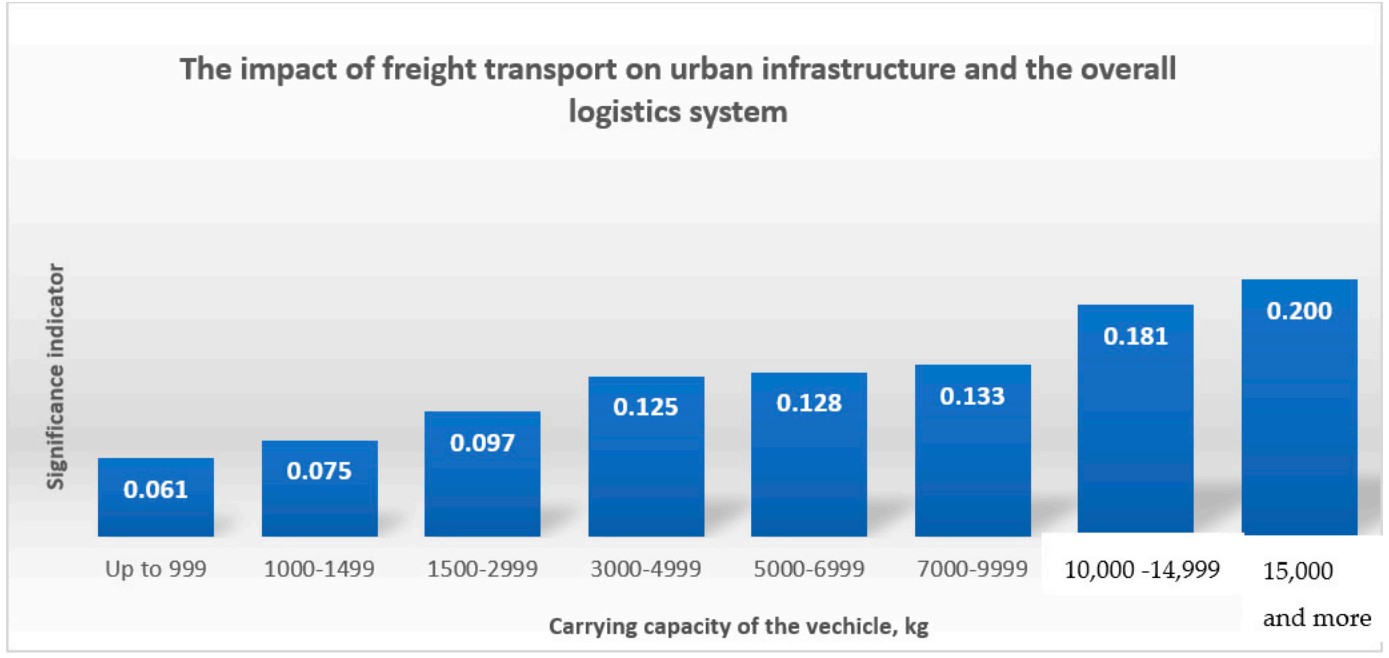

**Figure 3.** The impact of road vehicles on the infrastructure and the whole logistics system of a city.

The results of the study show that vehicles with a carrying capacity of up to 999 kg and 1000–1499 kg have the least impact on mobility and ecology in the city (significance indices of 0.061 and 0.075). However, vehicles with a carrying capacity of 15,000 kg or more are the most problematic and have the greatest impact on mobility and ecology in cities (importance index 0.200). At the same time, it can be assumed that vehicles up to 999 kg and 1000–1499 kg can be more easily replaced by electric vehicles.

Experts were asked how many companies would be willing to use small cargo terminals on the outskirts of a city. The vast majority of experts were in favor of such an initiative (seventy percent), ten percent were against it, and twenty percent of experts were likely to use it.

## 4. Discussion

From the analysis of scientific literature sources, several solutions for the optimization of cargo transport flows can be distinguished:

- The use and development of information technologies to manage, coordinate, and control information;
- Management efficiency through innovation in both urban regulations and environmental solutions;
- Placement of large terminals and small distribution centers in the countryside, taking into account the location of shopping centers and industrial districts;
- Infrastructure development using innovative solutions and information technologies;
- Collaboration, cooperation, and information and resource sharing among urban logistics participants.

The function of freight transport and the physical movement of goods from one place to another depends on the carrying capacity and regulation of the urban area. The problem is that these parameters are often not used if there is no cooperation between the public and private sectors and between business entities operating in one economic space in a certain territory in a city's logistics system.

Evaluating the purpose of this research, the chosen methodology, and the circumstances of the applicability of the obtained data, it can be said that the research covered all the research questions and tasks required by researchers. The study was focused on identifying possible optimization opportunities for city logistics by evaluating freight traffic and it was implemented.

In the studies conducted by researchers, it was not possible to find how the use of self-service terminals in peripheral logistics centers for small cargoes in city logistics, as is done when transporting small parcels, could affect freight traffic flows. Research has focused on the cooperation of logistics companies in the first or last mile and the use of information technology, ecological transport, and mobility, but has not examined the contribution of the shippers or receivers themselves to the planning and sustainable transportation process in urban logistics.

In summarizing the results of the previous research and the conducted research, it can be said that ideally, heavy transport should only be used for the direct transportation of bulky and heavy loads. In an urban logistics system, it should only be applied in the long or split first or last mile. However, due to the restriction of heavy freight transport in the central part of a city, transshipment points are needed for transshipment to commercial transport of smaller dimensions and carrying capacity.

Freight transport is least desirable in the central part of a city, because that is where it causes the most problems due to its dimensions, pollution, acceleration, and braking. Traffic flows are determined by the developed urban infrastructure, road capacity, traffic regulations, and the location of shopping centers, industrial centers, and logistics centers. In order for freight transport to move smoothly through the urban area, a suitable infrastructure is needed to optimize freight transport flows. This study revealed that the creation of a network of small self-service terminals on the outskirts of a city would benefit from such a function. Terminals of this type should be located on the outskirts of the central part of the city, taking into account the concentration and geographical location of business entities. Terminal functioning and servicing should be delegated to logistics companies with logistics warehouses outside of a city's territory. Business entities could transport small loads to the self-service terminals in the central part of the city through established transport channels.

Transport companies are looking for different ways to attract professional drivers, both from the local labor market and through migrants. In this case, drivers with category B driver's licenses would be sufficient for the first or last mile transportation in urban logistics, and heavy-duty vehicles driven by professional drivers would be better used to transport goods between the logistics center and self-service terminals. This should

reduce the need for such transport and at the same time help to control the lack of human resources in transport companies.

Freight transport, especially heavy freight transport, increases urban pollution and creates noise. In this paper, it is assumed that the use of self-service terminals for small loads in the periphery of cities would help reduce heavy traffic flows in cities. There would be a clear opportunity for the adoption of light-duty green vehicles, such as $CO_2$-free electric cars that would reduce pollution in cities. In addition, non-professional drivers can drive cars with a low carrying capacity up to 3.5 t. Such a vehicle could easily be driven not only by the driver of the transport company but also by the sender or recipient of the cargo. This would help solve another current problem: the lack of professional drivers.

The limitations of the study are given as follows. It can be said that the current study covers most of the important components of the urban logistics system, which are related to environmental impact and freight transport. The research aims to determine the general possibilities and trends in freight transport optimization by evaluating the individual factors that are related to this process; therefore, it can be said that there is an open possibility to additionally analyze each factor separately, analyze its significance, and determine the individual characteristics of the factor. Due to the limitations of the study, the potential vehicle exchange systems, information technologies, and other aspects are not detailed, but they are evaluated in the general context of the study.

## 5. Conclusions

The authors of scientific sources emphasize that city logistics has unique features, namely sensitivity to the impact on the environment, intensive delivery of cargo and parcels, and the need for continuous optimization of processes. Optimization processes are associated with routing, deployment of logistics centers, and the need for cargo consolidation–deconsolidation, with the possibility of using the PPP (public–private partnership) method in practice.

When analyzing the problems of urban logistics and freight transport flows, the authors of the scientific literature emphasize the challenges of the last mile related to the operation of the distribution system and make assumptions for developing the problem of the first mile, which is not widely analyzed. In summary, it can be stated that the distribution of freight transport flows in cities is mainly influenced by the geographical location of the city, the location of industrial and commercial places, and the locations of distribution centers in the territory and periphery. Furthermore, the mobility of freight transport in a city is influenced by road infrastructure, traffic regulations and restrictions, and parking capacity.

The expert survey method was used during this investigation; experts with sufficient education and professional practical experience were interviewed, taking into account the MCDM methodology, a sufficient number of experts were selected, and the data processing methodology used allowed us to state that the research results were reliable.

The results of the research showed that the greatest influences on the optimization of a city logistics system are the ban on entering the city center (indicator 0.2) and the ecological tax (0.172). The strategic role of the state and municipalities in the formation of rules for the use of a city logistics system and the promotion of insurance measures has been observed and evaluated. The study also showed that in city logistics, to optimize freight transport flows and improve the ecological climate, it is necessary to use cars weighing up to 999 kg, 1000–1499 kg, and 1500–2999 kg.

Taking into account the results of the study and modelling the possible actions, it is recommended to organize a network of self-service terminals to consolidate small amounts of cargo, which would solve the optimization problems of city logistics cargo flows raised during the analysis. This would allow shippers or consignees in the central part of the city to independently pick up or deliver goods to self-service terminals. This would make the transportation process easier to plan and prevent delays. It would also create a niche

for the targeted use of commercial electric vehicles in cities, helping to reduce pollution and noise.

A city's logistics system is mostly dependent on a city's infrastructure and traffic regulations. To optimize the flow of freight transport, a city's infrastructure must be constantly improved. It can be said that companies are ready to pick up or deliver small loads themselves to terminals on the outskirts of the city. Cooperation between institutions in consolidating or distributing cargo from terminals located on the outskirts of the city is important for customers. Additionally, companies take care of the optimization of cargo transportation flows by implementing information technologies to organize transport in the first or last mile.

The idea of self-service terminals on the outskirts of the city is attractive, but its novelty is questionable. Doubts are related to the fact that state support is not clear, its size is not clear, and it is not clear whether it will attract enough users.

A separate, additional topic of discussion, which this study reveals, is the competitive environment and its management possibilities. During commercial activities, representing clients may raise additional questions about cargo delivery times, order processing, consolidation, and waiting issues. This aspect needs to be further analyzed.

A separate area of additional research could be the details of the impact on the environment, assessing the elements of a city's logistics infrastructure, and the impact on them through emissions, noise, and accidents. However, it can be said that the current study is fully sufficient to create an initial reasonable picture of the situation confirmed by objective research methods, as well as the possibility of possible optimization methods.

An additional topic of research and to continue existing research can be the impact of state or municipal authorities based on the principle of a PPP (public–private partnership), their place and role in the development of possible cooperation systems, the establishment of new terminals, and issues of infrastructure development and use. This research topic is related to the current study, but has independent goals and requires a deeper analysis of specific issues.

**Author Contributions:** Conceptualization, N.B. and D.B.; methodology, D.B.; validation, N.B. and D.B.; formal analysis, N.B.; investigation, D.B.; resources, N.B.; data curation, N.B.; writing—original draft preparation, N.B.; writing—review and editing, N.B. and D.B.; visualization, N.B.; supervision, N.B.; project administration, N.B. and D.B.; funding acquisition, N.B. All authors have read and agreed to the published version of the manuscript.

**Funding:** This research was funded by Vilnius Gediminas Technical University.

**Institutional Review Board Statement:** Not applicable.

**Informed Consent Statement:** Not applicable.

**Data Availability Statement:** Not applicable.

**Acknowledgments:** The authors thank their colleague Šlajus Š. for their cooperation.

**Conflicts of Interest:** The authors declare no conflict of interest.

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
