# Peer review of "Solutions to the Problem of Freight Transport Flows in Urban Logistics"

_applsci, doi:10.3390/app13074214_

Round 1

Reviewer 1 Report

           1.     Are any information gaps in this study that haven't been addressed yet?

2.     You stated in line 205 that "All experts that were interviewed hold a university degree." What does the term "university degree" mean? Does it refer to a Bachelor's, Master's, or Philosophy degree?

3.     What selection criteria were used to choose the eight factors that were presented for the survey?

4.     Do you think that ten logistics experts is sufficient?

5.     It is preferable to write the equations down instead of using images of the equations at lines 313, 315, and 316.

          6.     It is important for the authors to identify any study limitations.

         7.     There was no mention of future study

Author Response

Response to Reviewer 1 Comments

First and foremost, thank you very much for your kind and valuable comments on our manuscript (2252130) with the title: “PROBLEMS SOLUTIONS OF FREIGHT TRANSPORT FLOWS IN URBAN LOGISTICS“. We appreciate all your time, insightful advice, and constructive comments that have helped us improve the paper for its publication. In revising the manuscript, we incorporated all your insightful comments and constructive suggestions; all necessary context changes are red-colored in the revision. The point-to-point responses are addressed as follows:

Point 1. Are any information gaps in this study that haven't been addressed yet?

Response: In part, this aspect is discussed when analyzing the issue of research limitations. However, evaluating the purpose of the research, the chosen methodology and the circumstances of the applicability of the obtained data, it can be said that the research covered all the research questions and tasks needed by the researchers. The research was focused on determining the possible optimization possibilities of city logistics by evaluating the traffic of freight transport and it was implemented.

Point 2. You stated in line 205 that "All experts that were interviewed hold a university degree." What does the term "university degree" mean? Does it refer to a Bachelor's, Master's, or Philosophy degree?

Response: Thanks for your insightful comment. All experts that were interviewed have a university degree - at least a bachelor's degree - and a combined average experience in logistics is almost 14 years.

Point 3. What selection criteria were used to choose the eight factors that were presented for the survey?

Response: This study covers most of the important components of the urban logistics system in terms of environmental impact and freight transport. The research aims to determine the general possibilities and trends of freight transport optimization by evaluating individual factors related to this process, therefore it can be said that there is an open possibility to additionally analyze each factor separately, its significance and the individual organizational characteristics of the factor.

Point 4. Do you think that ten logistics experts are sufficient?

Response: Thanks for your comment. The question is how many experts should be interviewed for the survey results to be sufficiently reliable. It has already been mentioned that all experts met the criteria of education (at least a bachelor's degree) and practical work experience. After evaluating the provisions and recommendations of Multi-Criteria Decision Making (MCDM) Methods and Concepts, their number is determined. The number of experts must be greater than the number of optional answers to the questions. In the case of our research, there are 10 experts, and the number of optional responses is 8, so it can be said that the number of experts is appropriate.

Point 5. It is preferable to write the equations down instead of using images of the equations at lines 313, 315, and 316.

Response: Thanks for your comment. The equations have been rewritten.

Point 6. It is important for the authors to identify any study limitations.

Response: Thanks for your comment. We presented study limitations in the Discussion section.

Point 7. There was no mention of future study.

Response: Thanks for your comment. The area of further research can be the detailing of the impact on the environment, evaluating the elements of the city's logistics infrastructure and the impact on them through emissions, noise, accidents.

We would like to thank you for your constructive help once again.

Reviewer 2 Report

The authors conducted a survey regarding freight transport flows in urban logistics. The abstract briefly names the aim, problem, method and results of the study, giving a good overview. The introduction gives a peek into the multidimensionality of the problem. During the introduction, the problem, the target group of the problem or the necessity of the paper and its research aim are not explicitly defined. The used references regarding the united nations are 20 years old. As we are now timely closer to 2050 and 2100 and the predictions got better over time, more recent references seem adequate. In addition, the reference for the number of people living in cities in 2014 (line 39) is missing and reference 4 is missing a year of publishing.

Then theoretical aspects present some aspects of urban freight transport and individual research within the field.  It remains unclear why are these individual approaches presented? How do they fit into the current state of research in this field? What other surveys were done in this field and how does yours differ from these? Also, fundamental terms should be shortly defined, as they are used within your work. A clear explanation of the stakeholder, their interests and aims in the matter should be given to further define the problem.

In the next subchapter, the research methodology is presented. The chosen method is a written survey. The aims of the survey should in my opinion be described before the method. The reasons for the choice of method seem rather thin. The suitability for scientific research is a basic condition, not an influencing factor for the final decision. How does the method contribute to solving the problem and how does it help to achieve the aims stated in this chapter? What are alternative methods and why are these not suited for the task at hand?  The number of interviewed people is given, also some background on them. However, restricting factors for the experts, for example nationality, are not given. The average (?) or the combined (as in total?) experience in logistics of all experts is not fully clear.  A clear statement whether ten participants in the survey are enough to yield scientifically sound results is missing.
The data evaluation methods are given and explained, but they are not reasoned or justified. Alternative options are not given.

The results named three questions and showed their results in great detail. While the results are given, it is unclear why these questions were asked and what other or whether only three questions were asked. Furthermore, it remains unclear what these results mean and how they should be interpreted. Do they make sense? How do they fit into the current research, do they match with the findings during the literature research? Or are they uncoupled from it? And how could that be explained? The figures 1 to 3 are rather large. This space could be used for further interpretation of the results.

The paper closes with a very brief discussion and conclusion. However, the discussion fails to discuss the assumptions, the method and its adequacy and applicability and the results. Several solutions to optimize the transport flows are presented. Why now and not in the second chapter? And how do you connect these solutions with your survey? Did you ask your participants about these solutions? Only one solution is highlighted and praised, but not discussed. The stakeholders and their preferences and for them necessary steps or challenges for implementation could be discussed. What are the pro and con arguments of this solution?

The brief conclusion highlights the chosen solution, but does not give a brief overview of the paper, as in what was done and why. In addition, a coherent, final conclusion is missing. There are only single sentences with their own content given, but no connection of arguments. An outlook into future research is missing, too. What could or will be done with the obtained results?

The language of the paper is understandable. The usage of tenses (especially in the introduction) should be revised. Some sentences are very long, which is very unusual for the English language. And some sentences do not make sense e.g. line 131 to line 133. 

Author Response

Response to Reviewer 2 Comments

First and foremost, thank you very much for your kind and valuable comments on our manuscript (2252130) with the title: “PROBLEMS SOLUTIONS OF FREIGHT TRANSPORT FLOWS IN URBAN LOGISTICS“. We appreciate all your time, insightful advice, and constructive comments that have helped us improve the paper for its publication. In revising the manuscript, we incorporated all your insightful comments and constructive suggestions; all necessary context changes are red-colored in the revision. The point-to-point responses are addressed as follows:

Point 1. The authors conducted a survey regarding freight transport flows in urban logistics. The abstract briefly names the aim, problem, method and results of the study, giving a good overview. The introduction gives a peek into the multidimensionality of the problem. During the introduction, the problem, the target group of the problem or the necessity of the paper and its research aim are not explicitly defined. The used references regarding the united nations are 20 years old. As we are now timely closer to 2050 and 2100 and the predictions got better over time, more recent references seem adequate. In addition, the reference for the number of people living in cities in 2014 (line 39) is missing and reference 4 is missing a year of publishing.

Response: Thanks for your insightful comment. In the introduction, we presented the problem, the target group of the problem and the purpose of the research. In the introduction, we presented the problem, the target group of the problem and the purpose of the research. Also, in this section we have provided more recent references to literature sources and supplemented them. Reference 4 has been supplemented.

Point 2. Then theoretical aspects present some aspects of urban freight transport and individual research within the field.  It remains unclear why are these individual approaches presented? How do they fit into the current state of research in this field? What other surveys were done in  this field and how does yours differ from these? Also, fundamental terms should be shortly defined, as they are used within your work. A clear explanation of the stakeholder, their interests and aims in the matter should be given to further define the problem.

Response: Thanks for your comment. In the theoretical part, we tried to examine as widely as possible what was emphasized when examining a similar topic. The analysis of literature sources helped to create the questionnaire as well. This was important to avoid duplication of studies. We have discovered an unexplored niche where there is a lot of focus on the collaboration of logistics companies in the first or last mile, green transport and mobility, but we have not explored the contribution of the shippers or consignees themselves to the planning and sustainable transport process in urban logistics. Our research was focused on identifying possible optimization opportunities for urban logistics by evaluating freight traffic and this was implemented.

Point 3. In the next subchapter, the research methodology is presented. The chosen method is a written survey. The aims of the survey should in my opinion be described before the method. The reasons for the choice of method seem rather thin. The suitability for scientific research is a basic condition, not an influencing factor for the final decision. How does the method contribute to solving the problem and how does it help to achieve the aims stated in this chapter? What are alternative methods and why are these not suited for the task at hand?  The number of interviewed people is given, also some background on them. However, restricting factors for the experts, for example nationality, are not given. The average (?) or the combined (as in total?) experience in logistics of all experts is not fully clear.  A clear statement whether ten participants in the survey are enough to yield scientifically sound results is missing.
The data evaluation methods are given and explained, but they are not reasoned or justified. Alternative options are not given.

Response: Thanks for your insightful comment. The purpose of the survey is moved before the interpretation of the method. All experts that were interviewed have a university degree - at least a bachelor's degree - and a combined average experience in logistics is almost 14 years. The respondents of the survey are experts in the Lithuanian logistics sector, heads of companies or company divisions.

The question is how many respondents should be interviewed for the survey results to be sufficiently reliable. It has already been mentioned that all experts met the criteria of education (at least a bachelor's degree) and practical work experience. After evaluating the provisions and recommendations of Multi-Criteria Decision Making (MCDM) Methods and Concepts, their number is determined. The number of experts must be greater than the number of optional answers to the questions. In the case of our research, there are 10 experts, and the number of optional responses is 8, so it can be said that the number of experts is appropriate.

Point 4. The results named three questions and showed their results in great detail. While the results are given, it is unclear why these questions were asked and what other or whether only three questions were asked. Furthermore, it remains unclear what these results mean and how they should be interpreted. Do they make sense? How do they fit into the current research, do they match with the findings during the literature research? Or are they uncoupled from it? And how could that be explained? The figures 1 to 3 are rather large. This space could be used for further interpretation of the results.

Response: Thanks for your insightful comment. These questions were asked as main ones. More questions were asked, such as how many years have you been working? what is your education what company do you work for? does your company contribute to the development of more efficient freight transportation? These responses are not the purpose of the study and therefore have not been commented on. The presented results are consistent with current research. We agree that our previous findings were not fully explained, so we have rewritten them.

Point 5. The paper closes with a very brief discussion and conclusion. However, the discussion fails to discuss the assumptions, the method and its adequacy and applicability and the results. Several solutions to optimize the transport flows are presented. Why now and not in the second chapter? And how do you connect these solutions with your survey? Did you ask your participants about these solutions? Only one solution is highlighted and praised, but not discussed. The stakeholders and their preferences and for them necessary steps or challenges for implementation could be discussed. What are the pro and con arguments of this solution?

Response: Thanks for your comment. We have reorganized and expanded the discussion section. We have added limitation and further actions.

Point 6. The brief conclusion highlights the chosen solution, but does not give a brief overview of the paper, as in what was done and why. In addition, a coherent, final conclusion is missing. There are only single sentences with their own content given, but no connection of arguments. An outlook into future research is missing, too. What could or will be done with the obtained results?

Response: Thanks for your comment. We have rewritten the Conclusions and resubmitted them. We also presented future research and what could be done with the results.

Point 7. The language of the paper is understandable. The usage of tenses (especially in the introduction) should be revised. Some sentences are very long, which is very unusual for the English language. And some sentences do not make sense e.g. line 131 to line 133. 

Response: Thanks for your insightful comment. We shortened some sentences. We have reworded the sentences from lines 131 to 133 for clarity. This text now looks like this:

„The examples presented show that freight transport is unevenly distributed in the city's logistics system. The use of freight transport in urban logistics is more on the capacity and legal regulation in the urban area. In the city logistics system, there are possible restrictions on the movement of freight and heavy vehicles: restrictions on driving on weekends and at appropriate hours, and restrictions on driving on streets that are in densely populated areas or near educational institutions. The load on the axle may also be limited, taking into account the existing state of the city's infrastructure, such as street surfaces, bridges, historical heritage, etc. City logistics is dominated by freight transport with a carrying capacity of up to 3.5 t, but the use of heavy-duty transport is also unavoidable, causing the most problems, especially during peak hours - possible traffic jams, noise, increased emissions due to braking - acceleration mode in traffic jams or controlled traffic light intersections.“

We would like to thank you for your constructive help once again.

Round 2

Reviewer 2 Report

Dear authors,

thank you for your changes to your paper, they improve it. However, some things remains unclear.

The authors conducted a survey regarding freight transport flows in urban logistics. The abstract briefly names the aim, problem, method and results of the study, giving a good overview. The introduction gives a peek into the multidimensionality of the problem. During the introduction, the stakeholders (target group and users, their interests and aims in the matter) of the problem are not explicitly defined. For the claims made in line 62 to 66 references are missing.

Then theoretical aspects present some aspects of urban freight transport and individual research within the field. At the end of the chapter, the approaches selected from the literature are combined to form the preferred solution. The chapter focuses very strongly on the chosen approach. Unfortunately, there is no explanation as to why this approach is focused on in the paper and what other solutions exist and why they are unsuitable. The opportunity to explain the answer options given in the survey and how they were chosen was unfortunately not taken, so these remain unclear.

In the next subchapter, the research methodology is presented. The chosen method is a written, quantitative survey. Its aimsare explained at the beginnning of the chapter. The reasons for the choice of method seem rather thin. The suitability for scientific research is a basic condition, not an influencing factor for the final decision. It remains unclear whether the average (?) or the total (combined ?) experience in logistics of all experts is named within the paper.

The data evaluation methods are given and explained, but they are not reasoned or justified. Alternative options are not given.

The results named three questions and showed their results in great detail. While the results are given, it remains unclear what these results mean and how they should be interpreted. How do they fit into the current research, do they match with the findings during the literature research? Or are they uncoupled from it? The figures 1 to 3 are rather large. This space could be used for further interpretation of the results. Figure 3 is depicted twice in the paper.

In line 457 it is tated, that 70 % of respondents are in favor of the presented initative, but only 20 % would support it: Where does the difference come from?

The paper closes with a very brief discussion and conclusion.

The discussion tries to explain the survey and its reasons. The chosen solution is better explained than before. The paper claims that its approach has not been discussed before and is a novelty. However, considering that DHL does something similar in e.g. Germany, with so called “parcel boxes”, the utter novelty of the approach is doubtful. Additional research areas are presented. In my opinion, they should be wihtin the conclusion.

The conclusion has greatly improved in comparison to the previous one. It highlights the chosen solution and gives a short overview over the paper. In your conclusion, you state that the “level of scientif study (...) is sufficiently broad.” Please elaborate this more, as the context of “broad” reamins unclear, as well as the relevance of this sentence for the conclusion. The last paragraph sums up the problem quite well. I think it is the collaboration of companies that is the problem with the approach rather than the attractiveness of the solution. State aid is a good point that can possibly be discussed in further research.

The language of the paper is understandable, but american and british english are mingled together. Some sentences are very long, which is very unusual for the English language. And some sentences do not make sense e.g. line 147 to line 150. 

Author Response

First and foremost, thank you very much for your kind and valuable comments on our manuscript (2252130) with the title: “PROBLEMS SOLUTIONS OF FREIGHT TRANSPORT FLOWS IN URBAN LOGISTICS “. We appreciate all your time, insightful advice, and constructive comments that have helped us improve the paper for its publication. In revising the manuscript, we incorporated all your insightful comments and constructive suggestions; all necessary context changes are red-colored in the revision. The point-to-point responses are addressed as follows:

Point 1. The authors conducted a survey regarding freight transport flows in urban logistics. The abstract briefly names the aim, problem, method and results of the study, giving a good overview. The introduction gives a peek into the multidimensionality of the problem. During the introduction, the stakeholders (target group and users, their interests and aims in the matter) of the problem are not explicitly defined. For the claims made in line 62 to 66 references are missing.

Response: Thanks for your insightful comment. In the introduction, we defined the stakeholders of the problem (the target group and users, their interests and goals in this matter). We added the missing links.

Point 2. Then theoretical aspects present some aspects of urban freight transport and individual research within the field. At the end of the chapter, the approaches selected from the literature are combined to form the preferred solution. The chapter focuses very strongly on the chosen approach. Unfortunately, there is no explanation as to why this approach is focused on in the paper and what other solutions exist and why they are unsuitable. The opportunity to explain the answer options given in the survey and how they were chosen was unfortunately not taken, so these remain unclear.

Response: Thanks for your comment. In the theoretical part, we tried to examine as widely as possible what was emphasized when examining a similar topic. The analysis of literature sources helped to create the questionnaire as well. We have discovered an unexplored niche where there is a lot of focus on the collaboration of logistics companies in the first or last mile, green transport and mobility, but we have not explored the contribution of the shippers or consignees themselves to the planning and sustainable transport process in urban logistics. Our research was focused on identifying possible optimization opportunities for urban logistics by evaluating freight traffic and this was implemented. In the survey, we provided only the essential answer options that reflect the problem we are dealing with.

Point 3. In the next subchapter, the research methodology is presented. The chosen method is a written, quantitative survey. Its aimsare explained at the beginnning of the chapter. The reasons for the choice of method seem rather thin. The suitability for scientific research is a basic condition, not an influencing factor for the final decision. It remains unclear whether the average (?) or the total (combined ?) experience in logistics of all experts is named within the paper. The data evaluation methods are given and explained, but they are not reasoned or justified. Alternative options are not given.

Response: Thanks for your insightful comment. Experts with an average logistics experience of almost 14 years were selected. An in-depth interview or other method can be an alternative to a survey, but a survey is more acceptable to both respondents and interviewers.

Point 4. The results named three questions and showed their results in great detail. While the results are given, it remains unclear what these results mean and how they should be interpreted. How do they fit into the current research, do they match with the findings during the literature research? Or are they uncoupled from it? The figures 1 to 3 are rather large. This space could be used for further interpretation of the results. Figure 3 is depicted twice in the paper. In line 457 it is tated, that 70 % of respondents are in favor of the presented initative, but only 20 % would support it: Where does the difference come from?

Response: Thanks for your insightful comment. We adjusted the results as much as we could.

The vast majority of experts were in favor of such an initiative (70 percent), 10 percent were against it, and 20 percent of experts were likely to use it.

Point 5. The paper closes with a very brief discussion and conclusion. The discussion tries to explain the survey and its reasons. The chosen solution is better explained than before. The paper claims that its approach has not been discussed before and is a novelty. However, considering that DHL does something similar in e.g. Germany, with so called “parcel boxes”, the utter novelty of the approach is doubtful. Additional research areas are presented. In my opinion, they should be wihtin the conclusion.

Response: Thanks for your comment. We've tweaked this section as much as we can. Additional areas of research are now presented in the conclusion.

Point 6. The conclusion has greatly improved in comparison to the previous one. It highlights the chosen solution and gives a short overview over the paper. In your conclusion, you state that the “level of scientif study (...) is sufficiently broad.” Please elaborate this more, as the context of “broad” reamins unclear, as well as the relevance of this sentence for the conclusion. The last paragraph sums up the problem quite well. I think it is the collaboration of companies that is the problem with the approach rather than the attractiveness of the solution. State aid is a good point that can possibly be discussed in further research.

Response: Thanks for your comment. We have supplemented the conclusion as follows:

“An additional topic of research and the continuity of existing research can be related to the impact of state or municipal authorities based on the principle of PPP (Public Private Partnership), their place and role in the development of possible cooperation systems, the establishment of new terminals, and issues of infrastructure development and use. This research topic is related to the current study, but has independent goals and requires a deeper analysis of specific issues.”

Point 7. The language of the paper is understandable, but american and british english are mingled together. Some sentences are very long, which is very unusual for the English language. And some sentences do not make sense e.g. line 147 to line 150. 

Response: Thanks for your insightful comment. We shortened some sentences. We tried to correct the language as best as we could. We have reworded the sentences from lines 147 to 150 for clarity.

We would like to thank you for your constructive help once again.

Round 3

Reviewer 2 Report

è The changed abstract no longer encompasses the content of the paper. Due to the abstract I would expect a literature research paper or a simulation paper. Please include a reference to your questionnaire as the heart of the scientific research in your paper.

è The designation of the stakeholders and their goals helps the reader to situate your research within the matter and complements your work.

è The added literature makes the conclusion at the end of the paper more coherent.

è The designation of the stakeholders and their goals as well as the added literature helps the reader to situate your research within the matter and makes the conclusion at the end of the paper more coherent.

Some of your sentences are very long (e.g. line 54 – 58(!)). The length makes them hard to understand. Please change that. 

Author Response

Thank you very much for your kind and valuable comments on our manuscript (2252130) with the title: “PROBLEMS SOLUTIONS OF FREIGHT TRANSPORT FLOWS IN URBAN LOGISTICS “. We appreciate all your time, insightful advice, and constructive comments that have helped us improve the paper for its publication. In revising the manuscript, we incorporated all your insightful comments and constructive suggestions; all necessary context changes are red-colored in the revision.

Point 1. The changed abstract no longer encompasses the content of the paper. Due to the abstract I would expect a literature research paper or a simulation paper. Please include a reference to your questionnaire as the heart of the scientific research in your paper.

Response: Thanks for your insightful comment. We have rewritten the abstract based on your suggestions.

Point 2. The designation of the stakeholders and their goals helps the reader to situate your research within the matter and complements your work.

Response: Thanks for your comment. We agree with your opinion.

Point 3. The added literature makes the conclusion at the end of the paper more coherent.

Response: Thanks for your comment. We agree with your opinion.

Point 4. The designation of the stakeholders and their goals as well as the added literature helps the reader to situate your research within the matter and makes the conclusion at the end of the paper more coherent.

Response: Thanks for your comment. We agree with your opinion.

Point 5. Some of your sentences are very long (e.g. line 54 – 58(!)). The length makes them hard to understand. Please change that. 

Response: Thanks for your comment. We have rewritten and shortened these sentences.

We would like to thank you for your constructive help once again.
